# Default Prediction with Industry-Specific Default Heterogeneity Indicators Based on the Forward Intensity Model

**Zhengfang Ni, Minghui Jiang * and Wentao Zhan**

School of Management, Harbin Institute of Technology, Harbin 150001, China; 17b910039@stu.hit.edu.cn (Z.N.); 22b910023@stu.hit.edu.cn (W.Z.)

*   Correspondence: jiangminghui@hit.edu.cn

**Abstract:** When predicting the defaults of a large number of samples in a region, this will be affected by industry default heterogeneity. To build a credit risk model that is more suitable for Chinese-listed firms, which have highly industry-specific default heterogeneity, we extend the forward intensity model to predict the defaults of Chinese-listed firms with information about the default heterogeneity of industries. Compared with the original model, we combine the Bayes approach with the forward intensity model to generate time-varying industry-specific default heterogeneity indicators. Our model can capture co-movements of different industries that cannot be observed based on the original forward intensity model so that the model can flexibly adjust the firm's PD according to the industry. In addition, we also consider the impact of default heterogeneity in other industries by studying the influence of the level and trends of other industries' default heterogeneity on a firm's credit risk. Finally, we compute PDs for 4476 firms from January 2001 to December 2019 for 36 prediction horizons. The extended model improves the prediction accuracy ratios both for the in-sample and out-of-sample firm's PDs for all 36 horizons. Almost all the accuracy ratios of the prediction horizons' PDs are increased by more than 6%. In addition, our model also reduces the gap between the aggregated PDs and the realized number of defaults. Our industry-specific default heterogeneity indicator is helpful to improve the model's performance, especially for predicting defaults in a large portfolio, which is of significance for credit risk management in China and other regions.

**Keywords:** forward intensity; industry; default heterogeneity; probability of default; Bayes approach

## 1. Introduction

### 1.1. Background and Purpose

Currently, credit risk management is an important topic for individual investors, banks, and financial regulators. To analyze the different structures of credit risk for multiple periods, credit risk models have developed to calculate the probability of default (PD) with concerning term structure. Firms' PDs have been employed by researchers and practitioners to do credit analysis on firms or large portfolios. Recently, researchers have found that industry-specific default heterogeneity exists among different industries. Traditional credit risk models, without information on industry-specific default heterogeneity, cannot meet the needs of financial regulators and obligors who loan to a large number of corporates. A credit risk model which does not consider industry-specific default heterogeneity cannot capture the co-movements of defaults in the industry. As the largest emerging market, China has a huge impact on the global economy. As the industry-specific default heterogeneity of Chinese-listed firms is complex, the prediction of Chinese-listed firms' PDs by various credit risk models is influenced by the industry's default heterogeneity.

Therefore, the purpose of this paper is to build a credit risk model that is more suitable for Chinese-listed firms than the traditional model. In addition, in order to solve the problem of default heterogeneity in different industries in China, this model needs to contain information on industrial default heterogeneity, so that it can adjust according to

the default heterogeneity of different industries when predicting the defaults of a large number of Chinese-listed firms, which makes the model more applicable. Such a model also has strong practicability when applied to the credit analysis of large portfolios, whether for China or other countries.

### 1.2. Research Objectives and Main Work

According to our research purpose, the main objectives of this paper are as follows: (1) We apply an advanced PD model to predict the PDs of Chinese-listed firms, which has rarely been used for Chinese firms before. Firstly, the model can realize dynamic measurement, which can automatically predict forward PDs with an update of the firm's risk factors. Secondly, the model should be able to predict the term structure of the PD and realize muti-period PD estimation. (2) In view of the default heterogeneity of different industries in China, we need to adjust the model to make the new model contain default heterogeneity information when predicting future defaults, so that our PDs have more accurate ratios. In addition, when predicting the defaults of a large number of Chinese-listed firms, the aggregated PDs should be closer to the realized default than the original model. As long as the research objectives are met, the model's practicability in the credit analysis of large portfolios can be improved.

Based on the above research objectives, this paper mainly addresses the following problems: (1) What PD model can realize the dynamic measurement of multi-period PDs for Chinese-listed firms? (2) How is it possible to improve the PD model to make it contain information on the default heterogeneity of different industries? (2) How is it possible to improve the accuracy ratios of multi-period PDs and reduce the gap between aggregated PDs and the realized number of defaults for multiple periods?

To solve the above problems, we develop a richer model based on the forward intensity approach proposed by Duan et al. [1], which has been applied to estimate the PDs of more than 73,000 listed firms worldwide. The forward intensity model can address the term structure effect and predict PDs for different horizons for a firm, which makes it an advanced credit risk measurement approach. Our model has two major differences compared with the original model. Firstly, we constructed time-varying industry-specific default heterogeneity indicators, which can reflect an unobserved part of the differences in defaults among different industries through a Bayesian approach based on the forward intensity model. The indicators we constructed can capture co-movements of defaults in all industries. Secondly, we consider the impact of co-movements of defaults both within industries and across industries when computing PDs. In fact, the situations of defaults in the industry and in the whole country both influence the PD of each listed firm to different degrees.

Because firms in different industries have different characteristics, we divided all Chinese-listed firms into 10 industries. Our extended model can measure the influence on different industries' firms. In our preliminary analysis, we found there was great heterogeneity of default among different industries. In this paper, we compute PDs according to default heterogeneity in different industries and improve the predictive ability of the forward intensity model.

### 1.3. Hypothesis, Novelties and Contributions

In this paper, Chinese-listed firms' PDs are estimated based on the forward intensity model. Comparing with the original model, we introduce the default heterogeneity indicator to characterize the default correlations within the industry. Therefore, the model in this paper has three hypotheses in total. Hypothesis (1): Firms' defaults and other exits obey the Poisson process. Since the forward intensity model is modified by the doubly stochastic Poisson intensity model, we keep this basic assumption. This also means that defaults and other exits do not affect the firms' covariates. Hypothesis (2): The firms' defaults are conditionally independent. This is also called CID hypothesis. The traditional default intensity model is usually built into the framework of the CID model. This

means that default correlations only depend on common factors, which are observable or potential. Hypothesis (3): Default heterogeneity causes default correlations within the industry. According to the CID hypothesis, we believe that the default correlations within the industry come from potential common factors, which can be described by industry default heterogeneity.

Compared with previous research, our novelties mainly include two aspects: (1) In recent research on the PD prediction of Chinese-listed companies, the forward intensity model, which has better performance, is rarely applied or modified by researchers compared with the structural model. In this paper, we fill this gap. (2) The original forward intensity model does not capture the co-movement of defaults within the industry when predicting the PDs of a large number of firms in a region. We propose a Bayesian approach to make the forward intensity model include information on industry default heterogeneity, so that the model can flexibly adjust the firm's PD according to industry.

Our main work is as follows. Firstly, we introduce the industry-specific default heterogeneity indicator into our model with a Bayesian approach and maximize the pseudo log-likelihood function to estimate the parameters of our Bayesian model. Then, default heterogeneity indicators for 10 industries are generated from January 2000 to December 2019. Secondly, we measure industry-specific default heterogeneity's influence on each Chinese-listed firm, both from its industry and from other industries, for all prediction horizons. In addition, our model also includes trends in industry-specific default heterogeneity indicators within the industry and among other industries. Due to default heterogeneity among different industries, we estimate four parameters of 10 industries for 36 prediction horizons in our second estimation, and, finally, we compute PDs from them based on forward intensity.

The main contributions of this paper are as follows: (1) We construct a credit risk measurement model considering industry-specific default heterogeneity to predict the PDs of Chinese-listed firms based on the forward intensity model. Including information on industry-specific default heterogeneity can make the model more practical in stress testing for an industry. (2) We introduce industry-specific default heterogeneity indicators into the forward intensity model. We capture co-movements in firms' defaults within the industry and among other industries for all prediction horizons. PDs are computed by considering the impact of default heterogeneity among different industries. (3) Compared with the original forward intensity model, our extended model improves the accuracy ratio and reduces the gap between aggregated PDs and the realized number of defaults for all prediction horizons.

This paper is organized as follows. In Section 2, we review the theoretical development of credit risk models and introduce recent research on industry-specific default heterogeneity. In Section 3, we extend the forward intensity model and introduce the industry-specific default heterogeneity indicator. This section also explains how we compute PDs. Section 4 gives details about how to estimate the parameters of our extended model. Section 5 presents the data and some preliminary analyses on defaults for all industries. In Section 6, we show the estimated values of the parameters of our model. We tested the model's performance for all horizons by comparing its predictive ability for PDs. Section 7 presents our conclusions and discusses future developments.

## 2. Literature Review

### 2.1. Credit Risk Models

Credit risk models that can compute PDs with term structure can be classified into two main categories: structural and reduced-form models. A structural model uses the structure of assets and liabilities to estimate the expected default frequency in the future. The earliest structural model was proposed by Merton [2] based on the famous Black–Scholes option pricing model. Later, the KMV company proposed the KMV model, based on the Merton model. Hillegeist et al. [3] believe that structural models have stronger predictive power than Z-scores. Because the structural model only relies on market information as

a predictive variable, it misses much information. Research usually combines structural models with other models, and recent examples are Zhang and Shi [4], Song et al. [5], and Zeng et al. [6].

In this paper, we extend the forward intensity model, which falls into the class of reduced-form models. Compared with structured models, reduced-form models make it possible to select risk factors freely, but early reduced-form models could not calculate PDs with term structure. The earliest reduced-form models date back to the 1960s. Altman and Edward [7] and Beaver William [8] only calculated the credit score using discriminant analysis. In 2007, Duffie et al. [9] proposed a doubly stochastic Poisson intensity model, making it possible to predict PD concerning its term structure. They addressed the term structure effect to realize multi-period prediction, which provides the theoretical basis for our study. In 2012, Duan et al. [1] proposed a forward intensity approach to make the doubly stochastic Poisson intensity model more practical. The forward intensity approach was implemented by maximizing the decomposable pseudo-likelihood function. They realized multi-period prediction using only the data available at the time they made the prediction from, providing the practical basis for this paper. Some researchers compute PDs based on the doubly stochastic Poisson intensity model or the forward intensity model. Recent examples are Hwang and Chu [10], Caporale et al. [11], Berent and Rejman [12], and Sigrist and Leuenberger [13].

### 2.2. Default Heterogeneity in Industries

Compared with structured models, the forward intensity approach includes more risk factors due to its broader assumptions. However, the original forward intensity approach computes PDs for a large sample of firms without considering default heterogeneity in industry and its influence. Recently, more and more researchers have studied industry-specific default heterogeneity among different industries. Bhimani et al. [14] computed PDs for 31,025 private enterprises in Portugal and found that industry and geographical location affect default. Dakovic et al. [15] constructed a generalized linear mixed model and modeled unobserved default heterogeneity in different industries, and found that a model considering unobserved heterogeneity had a higher accuracy ratio. Giesecke and Kim [16] developed a dynamic measurement method for systemic risk in the entire financial sector, which captures the impact of industry-specific risk factors on the timing of failure. Koopman et al. [17] proposed a high-dimensional, nonlinear, non-Gaussian dynamic factor model to decompose system default risk into (1) macro financial risk, (2) autonomous default dynamics, and (3) potential components of industry-specific effects. They found that about 35% of the variation in default rate was caused by industry-specific factors. Mensi et al. [18] investigated the short-term and long-term effects of FFR, VIX index, and crude oil price on credit risk in the US banking, financial service, and insurance sectors, which are quantile-dependent, and found that the short-term and long-term effects of risk factors are time-varying and heterogeneous under different credit market conditions. Gertler et al. [19] used the quasi-panel method to model the corporate loan default rates of four major economic industries. This model allowed the combination of long-term and short-term technologies at the same time to maintain a flexible unified framework, so as to capture heterogeneity among different industries, and found that significant default heterogeneity exists in industries. Lee [20] examined whether industry-level credit risk affects the yield spread of corporate bonds and used three types of industry risk variables: dilemma exposure measures, industry status, and product market competition. Evidence suggests that industry systemic risk does play an important role in explaining bond yield spreads.

This paper extends the forward intensity approach by including information on default heterogeneity in industries. We computed PDs with information on the default heterogeneity of all industries to make them more helpful for predicting defaults both for one firm and a large portfolio of firms.

## 3. Computed PDs by Industry-Specific Heterogeneity Indicators

### 3.1. Industry-Specific Default Heterogeneity Indicators

It is an effective method to construct different categories of samples' indicators as independent variables impacting credit risk. For example, Batrancea [21] constructed 10 ratios concerning financial performance to study their influence on bank assets and the liabilities of the most important 45 banks in Europe and Israel, the United States of America, and Canada, which provides a good basis for the research in this paper. In this paper, we extend the forward intensity model by constructing industry-specific heterogeneity indicators for all industries to compute PDs for multiple periods.

Duffie et al. [9] described defaults and other types of delisting excluding bankruptcy (other exit) as two independent doubly stochastic Poisson processes. For the $i$-th firm, default and other exit intensities can be denoted by $h_i(m,n)$ and $\overline{h}_i(m,n)$, respectively. The two intensities represent two average arrival rates during the interval $[n\tau,(n+1)\tau]$ with observation time point $m\tau$. Here, $\tau$ is set as one month, which means the basic time interval. Therefore, in a basic time interval, $\tau = 1/12$ and the default intensity is deterministic, so we can get the following probabilities:

$$\begin{aligned}
\mathrm{PD}_i(m,n) &= 1 - \exp(-h_i(m,n)\tau), \\
\mathrm{POE}_i(m,n) &= \exp(-h_i(m,n)\tau)\left[1 - \exp\left(-\overline{h}_i(m,n)\tau\right)\right], \\
\mathrm{PS}_i(m,n) &= \exp\left[-\left(h_i(m,n) + \overline{h}_i(m,n)\right)\tau\right].
\end{aligned} \tag{1}$$

Here, $\mathrm{PD}_i(m,n)$, $\mathrm{POE}_i(m,n)$, and $\mathrm{PS}_i(m,n)$ are the probabilities of default, other exit and survival, respectively, during the interval $[n\tau,(n+1)\tau]$ and observed at the time point $m\tau$. Obviously, $\mathrm{PD}_i(m,n) + \mathrm{POE}_i(m,n) + \mathrm{PS}_i(m,n) = 1$. For more details, readers are referred to Duan et al [1].

A lot of evidence shows that industry-specific default heterogeneity exists in different industries. In our preliminary analysis, Chinese-listed firms also show that there are different co-movements in different industries over time. Besides, a firm's default tendency changes with co-movements of defaults in the industry and in the whole economy. Section 5 shows the details of the preliminary analysis. To study unobserved co-movements of defaults in industries, we take an industry as a whole by averaging the default intensities of all the firms in the industry:

$$H_j(m,n) = \frac{\sum_{i=1}^{I_j(m)} h_i(m,n)}{I_j(m)}, \tag{2}$$

where $I_j(m)$ is the total number of surviving firms in the sample at the time point $m\tau$ in the $j$-th industry.

A Bayesian approach was often employed when modeling the firms' credit risk. Ni et al. [22] found that Bayesian estimation can be employed on the firm's default forward intensity to introduce default heterogeneity into the forward intensity model. In this paper, we take the industry's average default intensity, estimated by the forward intensity approach, as the average prior default intensity of the industry. According to the additivity of Poisson processes, the number of firm defaults in the $j$-th industry follows the Poisson distribution. If we take all firms in the $j$-th industry as a whole, the number of defaults for every firm in the $j$-th industry follows the Poisson distribution with the industry's average prior default intensity $H_j(m,n)$. According to the properties of the Poisson process, in a basic time interval, the conjugate prior distribution of $H_j(m,n)$ is a Gamma distribution expressed as $\Gamma\left(\alpha_j(m,n),\beta_j(m,n)\right)$, and

$$H_j(m,n) \sim \Gamma\left(\alpha_j(m,n),\beta_j(m,n)\right). \tag{3}$$

Then, the density function of $H_j(m,n)$ is

$$\pi\big(H_j(m,n)\big) = \frac{\beta_i(m,n)^{\alpha_i(m,n)}}{\Gamma(\alpha_i(m,n))} H_j(m,n)^{\alpha_i(m,n)-1} e^{-\beta_i(m,n)h_i(m,n)}. \tag{4}$$

Let $y_i(n)$ denote whether the $i$-th firm has a default during the interval $[n\tau, (n+1)\tau]$. When the observation time is after $(n+1)\tau$, the probability function of $y_i(n)$ is expressed as

$$p(y_i(n)) = \frac{h_i(m,n)^{y_i(n)}}{\Gamma(y_i(n)+1)} e^{-h_i(m,n)}. \tag{5}$$

Because default is a low-probability event, we assume a firm defaults at most once a month. Let $y_j(m,n)$ denote the number of defaults during the interval $[n\tau, (n+1)\tau]$ for all firms in the $j$-th industry which are surviving in the sample at time point $m\tau$. We then get the posterior distribution of $H_j(m,n)$ for the $j$-th industry when the observation time is after $(n+1)\tau$:

$$\pi\big(H_j(m,n) \mid y_j(m,n)\big) = \frac{p\big(y_j(m,n)|H_j(m,n)\big)\pi\big(H_j(m,n)\big)}{\int_0^{+\infty} p\big(y_j(m,n)|H_j(m,n)\big)\pi\big(H_j(m,n)\big)dH_j(m,n)},$$

$$= \frac{\left[\frac{H_j(m,n)^{\sum_{i=1}^{I_j(m)} y_i(n)} e^{-I_j(m)H_j(m,n)}}{\Pi_{i=1}^{I_j(m)}\Gamma(y_i(n)+1)}\right]\left[\frac{\beta_j(m,n)^{\alpha_j(m,n)}}{\Gamma(\alpha_j(m,n))} H_j(m,n)^{\alpha_j(m,n)-1} e^{-\beta_j(m,n)H_j(m,n)}\right]}{\int_0^{+\infty}\left[\frac{H_j(m,n)^{\sum_{i=1}^{I_j(m)} y_i(n)} e^{-I_j(m)H_j(m,n)}}{\Pi_{i=1}^{I_j(m)}\Gamma(y_i(n)+1)}\right]\left[\frac{\beta_j(m,n)^{\alpha_j(m,n)}}{\Gamma(\alpha_j(m,n))} H_j(m,n)^{\alpha_j(m,n)-1} e^{-\beta_j(m,n)H_j(m,n)}\right]dH_j(m,n)}$$

$$= \frac{\big(I_j(m)+\beta_i(m,n)\big)^{\alpha_i(m,n)+y_j(m,n)}}{\Gamma(\alpha_i(m,n)+y_j(m,n))} H_j(m,n)^{\alpha_i(m,n)+y_j(m,n)-1} e^{-(I_j(m)+\beta_i(m,n))H_j(m,n)}. \tag{6}$$

Let $\hat{H}_j(m,n)$ be the parameter of the posterior intensity of $H_j(m,n)$, then

$$\hat{H}_j(m,n) \sim \Gamma\big(\alpha_j(m,n) + y_j(m,n), \beta_j(m,n) + I_j(m)\big). \tag{7}$$

According to the properties of the Gamma distribution, during the period $[n\tau, (n+1)\tau]$, the mean values of the prior and posterior default intensities can be expressed correspondingly as

$$E\big(H_j(m,n)\big) = \frac{\alpha_j(m,n)}{\beta_j(m,n)} = \tau H_j(m,n),$$
$$E\big(\hat{H}_j(m,n)\big) = \frac{\alpha_j(m,n)+y_j(m,n)}{\beta_j(m,n)+I_j(m)} = \tau \hat{H}_j(m,n). \tag{8}$$

Combining the above formulas to eliminate $\alpha_j(m,n)$, we have a relationship between $H_j(m,n)$ and $\hat{H}_j(m,n)$:

$$\hat{H}_j(m,n) = \frac{\beta_j(m,n)H_j(m,n) + \frac{y_j(m,n)}{\tau}}{\beta_j(m,n) + I_j(m)}. \tag{9}$$

According to the properties of conjugate distributions, the higher $\beta_i(m,n)$, the more confidence we have in $H_j(m,n)$. Specially, as $\beta_i(m,n)$ approaches positive infinity, $\hat{H}_j(m,n) = H_j(m,n)$, which means default intensity computed by the forward intensity approach completely dominates posterior default intensities. On the contrary, if $\beta_i(m,n) = 0$, $\hat{H}_j(m,n) = \frac{y_j(n)}{\tau I_j(m)}$ means that the posterior default intensity depends on the data we observe, which is the average number of defaults of all firms in the $j$-th industry during the $n$-th month. If we combine the above formulas:

$$\hat{H}_j(m,n) = \frac{\beta_j(m,n) + I_j(m)\frac{y_j(m,n)}{\tau\sum_{i=1}^{I_j(m)} h_i(m,n)}}{\beta_j(m,n) + I_j(m)} H_j(m,n). \tag{10}$$

The above equation shows the relationship between the average posterior intensity and the average prior intensity for all the *j*-th industry's firms. Here, $\tau \sum_{i=1}^{I_j(m)} h_i(m,n)$ is the expectation of the number of defaults in the *j*-th industry in month *n*, which is estimated using the forward intensity approach, while $y_j(m,n)$ is actual number of defaults in the *n*-th month in the same sample from this industry. If $\tau \sum_{i=1}^{I_j(m)} h_i(m,n) > y_j(m,n)$, it means that the original model overestimates the default intensity of the whole industry in the *n*-th month, then $\hat{H}_j(m,n) < H_j(m,n)$. On the contrary, if $\tau \sum_{i=1}^{I_j(m)} h_i(m,n) < y_j(m,n)$, it means that the original model underestimates the default intensity of the whole industry in the *n*-th month, then $\hat{H}_j(m,n) < H_j(m,n)$. Let the ratio of the posterior default intensity to the prior default intensity be the industry-specific default heterogeneity indicator, which reflects the extent to which the total number of defaults in the industry exceeds or falls below the aggregated prior PDs.

$$Z_j(m,n) = \frac{\hat{H}_j(m,n)}{H_j(m,n)} = \frac{\beta_j(m,n) + I_j(m)\frac{y_j(m,n)}{\tau \sum_{i=1}^{I_j(m)} h_i(m,n)}}{\beta_j(m,n) + I_j(m)}. \tag{11}$$

Note that the industry-specific default heterogeneity indicator is time-varying, and depends on the difference between the total number of firm defaults in the *n*-th month and the expectation of the number of defaults estimated by the original model in the same industry in the same month. It can capture the co-movements of defaults in the industry. Because the original model does not consider industry-specific default heterogeneity, it can only capture less information on the time dynamics in the industry or the whole economy by including observable common factors in the input variables. Our industry-specific default heterogeneity indicator is calculated by comparing the posterior and prior default intensity in the industry, which can capture an unobserved part of the co-movements of defaults in industries. For example, when the average posterior default intensity is much greater than the average prior default intensity of an industry, we can know a co-rise of defaults in the industry has not been captured by the original model. If we ignore this information, we will underestimate the firms' PDs when the default cluster occurs in this industry. Besides, we can also measure the average tendency to default in different industries over different time periods due to its time-varying nature. On the other hand, for industries with low credit risk, due to fewer default events within the industry, the number of realized defaults for different default prediction horizons will be far less than the aggregated PDs for all firms within the industry. According to Formula (11), the industry-specific default heterogeneity indicator $Z_j(m,n)$ will be less than 1. If a firm's credit risk is very dependent on the industry-specific default heterogeneity indicator for its industry, when we estimate firms' PDs in this industry, the new default forward intensity will be obviously lower than the original default forward intensity, which means the new PDs are lower. Therefore, when we estimate PDs, we will introduce default heterogeneity in the industries into the forward intensity function. Duan and Miao [23] found that short-term PDs can capture more co-movements in credit risk. Therefore, we calculated the shortest horizon's industry-specific default heterogeneity indicators to capture most co-movements by set $n = m$. Then, $h_i(m,m)$ denotes the *i*-th firm's default intensity during the *m*-th month observed on the first day of the *m*-th month. We assume that for firms in the same industry the posterior 1-month PDs share the same confidence in the 1-month prior PD. Then, we replace $\beta_j(m,m)$ and $Z_j(m,m)$ by $\beta_j(1)$ and $Z_j(m)$ respectively.

$$Z_j(m) = \frac{\beta_j(1) + I_j(m)\frac{y_j(m,m)}{\tau \sum_{i=1}^{I_j(n)} h_i(m,m)}}{\beta_j(1) + I_j(m)}. \tag{12}$$

To predict forward PDs in the future, we assume that an industry maintains a similar industry default heterogeneity during close periods. We calculate the most recent available

industry-specific default heterogeneity indicators to represent co-movements of defaults in industries in the current month. We define the industry-specific posterior default intensity of the $m$-th month in terms of the prior default intensity multiplied by the most recent available industry-specific default heterogeneity indicator:

$$\hat{h}_{ij}(m) = Z_j(m-1)h_i(m,m),$$ (13)

Here, we use 1-month-horizon prior default intensity and an industry-specific default heterogeneity indicator to make $\hat{h}_{ij}(m)$ as close as possible to the current situation. When we observe at the first day of the $m$-th month, $Z_j(m,m)$ is unknown. Then, the industry-specific posterior default intensity is as follows:

$$\hat{h}_{ij}(m) = \frac{\beta_j(1) + I_j(m-1)\frac{y_j(m-1,m-1)}{\tau \sum_{i=1}^{I_j(m)} h_i(m-1,m-1)}}{\beta_j(1) + I_j(m-1)} h_i(m,m).$$ (14)

The 1-month industry-specific posterior default intensity of every firm is calculated taking industry-specific default heterogeneity into account. In addition, if the PDs of an industry are aggregated, the systematic credit risk of the industry can also be estimated, which can be applied for stress testing of the industry's credit risk. This way of calculating portfolio credit risk by aggregating every firm's PD is a bottom-up method. However, this industry-specific posterior default intensity cannot be regarded as our final calibrated default intensity. This is because a firm's default can not only be influenced by the industry the firm is in, but may also be influenced by other industries. Therefore, if we calibrate a firm's PD, we need to take all industries' default situations into account. But we need industry-specific posterior default intensity to estimate $\beta_j(1)$, which will be introduced in the next section. With $\beta_j(1)$, we can calculate $Z_j(m)$ for all past months, which reflects the average level of default heterogeneity of the $j$-th industry in the $m$-th month.

*3.2. PDs for All Horizons*

If the default heterogeneity of other industries is not taken into account, the impact of co-movement of the whole economy's defaults will not be adequately considered. Thus, we constructed a new variable, $Z_{j,other}(m)$, denoting the weighted average industry-specific default heterogeneity of other firms:

$$Z_{j,other}(m) = \frac{\sum_{s=1}^{10} Z_s(m)I_s(m) - Z_j(m)I_j(m)}{\sum_{s=1}^{10} I_s(m) - I_j(m)}.$$ (15)

Here, $Z_j(m)$ and $Z_{j,other}(m)$ both contribute to our calibrated default intensity. On the other hand, when we compute the longer prediction horizons' PDs, it is not enough to just consider the current value of industry-specific default heterogeneity. Duan et al [1] found trends in some input variables are helpful to predict PDs. We also take the trends in our industry-specific default heterogeneity indicators into account. Trends are calculated using the difference between current value and the average value over the past period of time. To be consistent, we take the average value of the past 12 months following Duan et al [1]. Because the default intensity function is a proportional-hazards form of the original forward intensity approach, our forward default intensity function is as follows:

$$\hat{h}_i(m,m+l-1) = \exp(X_j(m)\gamma_j(l))h_i(m,m+l-1).$$ (16)

Here, $\gamma_j(l)$ is a column vector and $X_j(m)$ is a row vector:

$$\gamma_j(l) = \left[\gamma_{j1}(l), \gamma_{j2}(l), \gamma_{j3}(l), \gamma_{j4}(l)\right]^T,$$
$$X_j(m) = \left[Z_j(m-1), \, TZ_j(m-1), Z_{j,other}(m-1), TZ_{j,other}(m-1)\right],$$ (17)

where $X_j(m)$ is the industry-specific default heterogeneity indicator of the $j$-th industry. Here, $TZ_j(m-1)$ and $TZ_{j,other}(m-1)$ are the trends of $Z_j(m-1)$ and $Z_{j,other}(m-1)$, respectively. The next section will describe how we estimate $\gamma_j(l)$ for all the prediction horizons. As long as we estimate $\gamma_j(l)$, we can calibrate the default intensity estimated by the forward intensity approach to compute $\hat{h}_i(m,n)$, which considers co-movements in the industry's credit risk. With $\hat{h}_i(m,n)$, we can compute cumulative POEs, PDs for different prediction horizons. Firstly, we compute conditional PDs and POEs by $\hat{h}_i(m,n)$ and $\bar{h}_i(m,n)$, where $\bar{h}_i(m,n)$ is calculated by the original models. Then, we can compute forward PDs, which are conditional PDs timed by probabilities that the firm survives between the predicting month and the observed month. Finally, we can compute cumulative PDs by cumulating forward PDs. For more details of computing PDs, readers may also refer to Duan et al. [1].

In this paper, we compute cumulative PDs for horizons from 1 month to 36 months and evaluate the prediction performance of our PDs computed with industry-specific default heterogeneity and PDs estimated by the original model in Section 6.

## 4. Pseudo-Likelihood Functions

Firstly, we employ the forward intensity model to calculate default forward intensity. Then, let $\tau_i$ and $\bar{\tau}_i$ be the months after the $(m-1)$-th months in which the default and other exit occur, respectively, for the $i$-th firm. We assume the firms' defaults are conditionally independent, and the 1-month horizon pseudo-likelihood function of the $j$-th industry's posterior probability is expressed as:

$$\mathrm{P}^{posterior}_{l=1}\left(\beta_j(1),\tau_D,\tau_{OE},h\right) = \prod_{m=1}^{M}\prod_{j=1}^{J}\prod_{i=1}^{I_j(m)}\hat{P}^{posterior}_{j,l=1}\left(\beta_j(1),\tau_i,\bar{\tau}_i,h_i(m,m)\right) \quad (18)$$

where $M$ denotes the last month of the sample, $I$ is the number of firms in the sample, $J$ is the number of industries in the sample, $\hat{P}_l(\beta_l,\tau_{Di},\tau_{OEi},h_i(m))$ is a probability depending on the actual status of the $i$-th firm during month $m$, and $h_i(m)$ is default intensity estimated by the forward intensity model on the first day of month $m$ for the $i$-th firm. Then,

$$\begin{aligned}
\hat{P}^{posterior}_{j,l=1}\left(\beta_j(1)\,\tau_i,\bar{\tau}_i,h_i(m,m)\right) &= 1_{\{t_{0i}\leq m,\min(\tau_i,\bar{\tau}_i)>m\}}\mathrm{P}(t_{0i}\leq m,\min(\tau_i,\bar{\tau}_i)>m) \\
&\quad +1_{\{t_{0i}<m,\tau_i\leq\bar{\tau}_i,\tau_i=m\}}\mathrm{P}(t_{0i}\leq m,\bar{\tau}_i\leq\bar{\tau}_i,\tau_i=m) \\
&\quad +1_{\{t_{0i}<m,\bar{\tau}_i\leq\tau_i,\bar{\tau}_i=m\}}\mathrm{P}(t_{0i}\leq m,\bar{\tau}_i\leq\tau_i,\bar{\tau}_i=m)+1_{\{t_{0i}>m\}}\mathrm{P}(t_{0i}>m) \\
&\quad +1_{\{\min(\tau_i,\bar{\tau}_i)<m\}}\mathrm{P}(\min(\tau_i,\bar{\tau}_i)<m),
\end{aligned} \quad (19)$$

where $t_{0i}$ is the first month for the $i$-th firm. The above formula is like the overlapped pseudo-likelihood function proposed by Duan et al. [1]. Here, the first term is the probability the firm survives during month $m$. The second term is the probability the firm defaults during month $m$. The third term is the probability the firm has another exit event during month $m$. The last two terms are the situations where the firm has not entered the sample and the firm has already exited from the market. We introduced Formula (14) into the overlapped pseudo-likelihood function:

$$
\hat{P}^{\text{posterior}}_{j,l=1}\left(\beta_j(1), \tau_i, \overline{\tau}_i, h_i(m,m)\right)
$$

$$
= 1_{\{t_{0i}\leq m, \min(\tau_i,\overline{\tau}_i)>m\}} \times \exp\left\{-\tau\left[\frac{\beta_j(1)+I_j(m-1)\frac{y_j(m-1,m-1)}{I_j(m)}}{\beta_j(1)+I_j(m-1)}h_i(m,m)+\overline{h}_i(m,m)\right]\right\}
$$

$$
+1_{\{t_{0i}\leq m, \tau_i\leq\overline{\tau}_i, \tau_i=m\}}\left\{1-\exp\left[-\tau\frac{\beta_j(1)+I_j(m-1)\frac{y_j(m-1,m-1)}{I_j(m)}}{\beta_j(1)+I_j(m-1)}h_i(m,m)\right]\right\}
$$

$$
+1_{\{t_{0i}\leq m, \overline{\tau}_i\leq\tau_i, \overline{\tau}_i=m\}}\left\{-\exp\left[-\tau\overline{h}_i(m,\mathrm{m})\right]\right\} \times \exp\left[-\tau\frac{\beta_j(1)+I_j(m-1)\frac{y_j(m-1,m-1)}{I_j(m)}}{\beta_j(1)+I_j(m-1)}h_i(m,\mathrm{m})\right]
$$

$$
+1_{\{t_{0i}>m\}} + 1_{\{\min(\tau_i,\overline{\tau}_i)<m\}}. \tag{20}
$$

Then, we keep the terms associated with $\beta_j(1)$:

$$
\hat{P}^{\text{posterior},\beta}_{j,l=1}\left(\beta_j(1), \tau_i, \overline{\tau}_i, h_i(m,m)\right)
$$

$$
= 1_{\{t_{0i}\leq m, \min(\tau_i,\overline{\tau}_i)>m\}} \times \exp\left\{-\tau\left[\frac{\beta_j(1)+I_j(m-1)\frac{y_j(m-1,m-1)}{I_j(m)}}{\beta_j(1)+I_j(m-1)}h_i(m,m)\right]\right\}
$$

$$
+1_{\{t_{0i}\leq m, \tau_i\leq\overline{\tau}_i, \tau_i=m\}}\left\{1-\exp\left[-\tau\frac{\beta_j(1)+I_j(m-1)\frac{y_j(m-1,m-1)}{I_j(m)}}{\beta_j(1)+I_j(m-1)}h_i(m,m)\right]\right\}
$$

$$
+1_{\{t_{0i}\leq m, \overline{\tau}_i\leq\tau_i, \overline{\tau}_i=m\}}\exp\left[-\tau\frac{\beta_j(1)+I_j(m-1)\frac{y_j(m-1,m-1)}{I_j(m)}}{\beta_j(1)+I_j(m-1)}h_i(m,m)\right]
$$

$$
+1_{\{t_{0i}>m\}} + 1_{\{\min(\tau_i,\overline{\tau}_i)<m\}}. \tag{21}
$$

Appendix A shows how to maximize $L^{\text{posterior}}_{l=1}\left(\beta_j(1), \tau_D, \tau_{OE}, h\right)$ and estimate $\beta_j(1)$. With $\beta_j(1)$, we can compute all firms' PDs and calculate the trends and current values of our industry-specific default heterogeneity indicators. The industry-specific default heterogeneity indicators of all industries are then calculated. The pseudo-likelihood function of new default intensity with industry-specific default heterogeneity indicators is expressed as:

$$
P_l(\gamma, \tau_D, \tau_{OE}, X, h) = \prod_{m=1}^{M-1}\prod_{j=1}^{J}\prod_{i=1}^{I_j(m)}\hat{P}_{j,l}\left(\gamma_{j1}, \gamma_{j2}, \gamma_{j3}, \gamma_{j4}, \tau_i, \overline{\tau}_i, X_j(m), h_i(m, m+l-1)\right) \tag{22}
$$

where

$$\hat{P}_{j,l}\left(\gamma_{j1}, \gamma_{j2}, \gamma_{j3}, \gamma_{j4}, \tau_i, \overline{\tau}_i, X_j(m), h_i(m, +l-1)\right)$$

$$= 1_{\{t_{0i} \le m, \min(\tau_i, \overline{\tau}_i) \ge m+l\}} \exp\left(-\tau \left\{ \sum_{k=0}^{l-1} \left\{ \exp\left[X_j(m)\gamma_j(k+1)\right] h_i(m, m+k) + \overline{h}_i(m, m+k) \right\} \right\}\right)$$

$$+ 1_{\{t_{0i} \le m, \tau_i \le \overline{\tau}_i, \tau_i < m+l\}} \exp\left(-\tau \left\{ \sum_{k=0}^{\tau_i-m-1} \left\{ \exp\left[X_j(m)\gamma_j(k+1)\right] h_i(m, m+k) + \overline{h}_i(m, m+j) \right\} \right\}\right)$$

$$\times \left\{ 1 - \exp\left[ -\tau \exp\left[X_j(m)\gamma_j(\tau_i-m+1)\right] h_i(m, \tau_i) \right] \right\}$$

$$+ 1_{\{t_{0i} \le m, \overline{\tau}_i \le \tau_i, \overline{\tau}_i < m+l\}} \exp\left(-\tau \left\{ \sum_{k=0}^{\overline{\tau}_i-m-1} \left\{ \exp\left[X_j(m)\gamma_j(k+1)\right] h_i(m, m+k) + \overline{h}_i(m, m+k) \right\} \right\}\right)$$

$$\times \left\{ 1 - \exp\left[ -\tau \overline{h}_i(m, \overline{\tau}_i) \right] \right\} \exp\left\{ -\tau \exp\left[X_j(m)\gamma_j(\tau_i-m+1)\right] h_i(m, \tau_i) \right\}$$

$$+ 1_{\{t_{0i} > m\}} + 1_{\{\min(\tau_i, \overline{\tau}_i) < m\}}.$$

(23)

Here, the first term is the probability that a firm survives during $l$ months from the $m$-th month. The second term is the probability that a firm defaults during $l$ months from the $m$-th month. The third term is the probability that a firm has another exit during $l$ months from the $m$-th month. The last two terms are the situations that a firm has not entered the sample and that a firm has already exited the market. Similarly, we keep the terms associated with $\gamma$:

$$\hat{P}_{j,l}^{\gamma}\left(\gamma_{j1}, \gamma_{j2}, \gamma_{j3}, \gamma_{j4}, \tau_i, \overline{\tau}_i, X_j(m), h_i(m, +l-1)\right)$$

$$= 1_{\{t_{0i} \le m, \min(\tau_i, \overline{\tau}_i) \ge m+l\}} \exp\left[ -\tau \sum_{k=0}^{l-1} \exp\left[X_j(m)\gamma_j(k+1)\right] h_i(m, m+k) \right]$$

$$+ 1_{\{t_{0i} \le m, \tau_i \le \overline{\tau}_i, \tau_i < m+l\}} \exp\left[ -\tau \sum_{k=0}^{\tau_i-m-1} \exp\left[X_j(m)\gamma_j(k+1)\right] h_i(m, m+k) \right]$$

$$\times \left\{ 1 - \exp\left[ -\tau \exp\left[X_j(m)\gamma_j(\tau_i-m+1)\right] h_i(m, \tau_i) \right] \right\}$$

$$+ 1_{\{t_{0i} \le m, \overline{\tau}_i \le \tau_i, \overline{\tau}_i < m+l\}} \exp\left[ -\tau \sum_{k=0}^{\overline{\tau}_i-m-1} \exp\left[X_j(m)\gamma_j(k+1)\right] h_i(m, m+k) \right]$$

$$\times \exp\left[ -\tau \exp\left[X_j(m)\gamma_j(\tau_i-m+1)\right] h_i(m, \tau_i) \right]$$

$$+ 1_{\{t_{0i} > m\}} + 1_{\{\min(\tau_i, \overline{\tau}_i) < m\}}.$$

(24)

Details about how to maximize $P_l(\gamma, \tau_D, \tau_{OE}, X, h)$ and to estimate $\gamma$ are also introduced in Appendix A.

## 5. Data and Preliminary Analysis

### 5.1. Data

Our data consist of firm-specific variables, firms' events information, and common factors obtained from the NUS-CRI database. Default events are defined using the standards of CRI: (1) bankruptcy filing; (2) a missed or delayed payment; or (3) debt restructuring/distressed exchange. Our data set is all Chinese-listed firms from 1991 to 2020. In our sample, we used firm-specific and common variables for the first available day of each month from January 2000 to December 2019 to estimate our models' parameters. To test the out-of-sample performance, the data were divided into the experimental group and the evaluation group at a ratio of 5 to 1, randomly. There were in total 3747 and 729 firms in the experimental group and the evaluation group, respectively.

To compare the model, we construct the forward intensity function with the same variables used by Duan et al. [1]. There are four common factors: stock index return, interest rate, financial aggregated DTD, and non-financial aggregated DTD. Here, aggregated DTD is the median DTD of all financial or non-financial Chinese-listed firms. DTD is calculated by an adjustment method provided by Duan and Wang [24]. The firm-specific variables include DTD, CASH/TA, CA/CL, NI/TA, SIZE, M/B, and SIGMA. Duan et al. [1] found it helpful for estimating default to consider the trends in some firm-specific variables, which are the differences in value between the current values and the 1-year mean values of the

variables. Therefore, DTD, CASH/TA, CA/CL, NI/TA, and size are calculated for the trend and level. Here, the level is the 1-year mean value of the variable.

### 5.2. Preliminary Analysis

According to the NUS-CRI database, Chinese-listed firms are divided into 10 industries: basic materials, communications, consumer (cyclical), consumer (non-cyclical), diversified, energy, financial, industrial, technology, and utilities. Figure 1 shows the number of all industry defaults per year from January 2000 to December 2019. Figure 2 shows the frequency of all industry defaults per year in the same period. Because there are many industries, we show five industries with more defaults and five industries with fewer defaults in two figures, respectively.

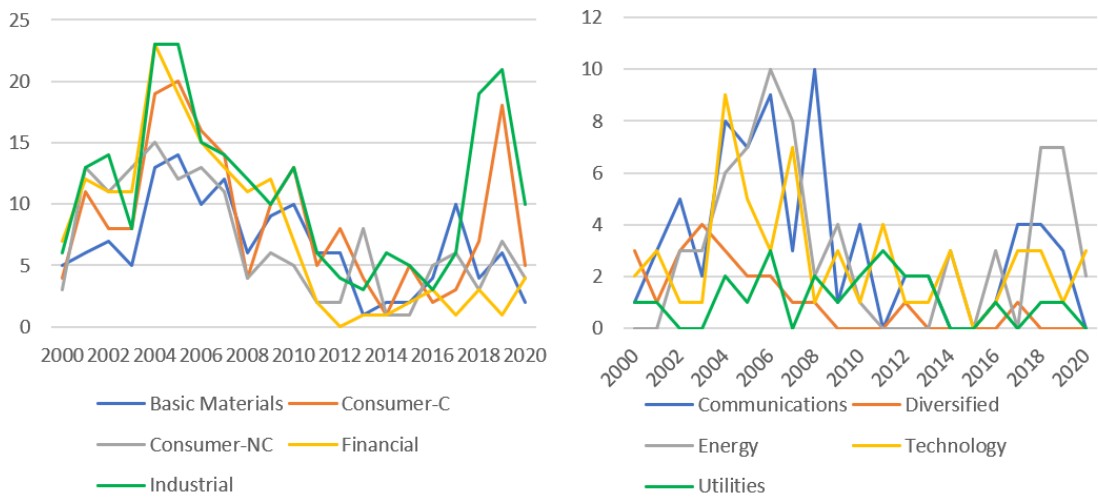

**Figure 1.** The number of all industry defaults per year. Source: NUS-CRI database.

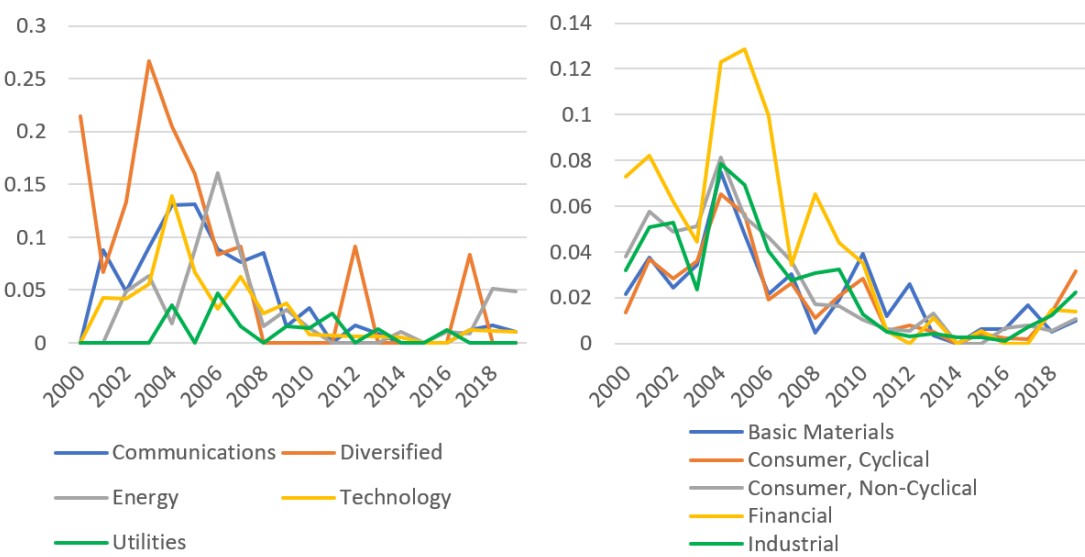

**Figure 2.** The frequency of all industry defaults per year. Source: NUS-CRI database.

We find that there are different fluctuations of default frequency in different industries. In terms of the overall default ratio, diversified firms have a higher default frequency than average. Comparatively, PDs in the utilities industry are very low. In addition, there are differences in time span and degree of fluctuation in different industries. From 2003 to 2007, for example, default frequency was high in the financial and communications industries. Compared with the energy industry, the time span of default clustering is between 2004 and

2008. We believe that corporate defaults are not only related to the overall default ratio and trend in the region but also related to the overall default ratio and trends in the industry.

Currently, credit risk models are able to measure a firm's PDs for multiple periods. By aggregating the PDs of all firms, the PD of the whole industry or region can be obtained. Therefore, credit risk models can also be employed to stress test. However, if a credit risk model does not contain information on industry-specific default heterogeneity, it cannot capture unobserved co-movements of defaults in the industry. Our model considers different co-movements of defaults in all industries. The prediction performance for aggregated PDs calculated by a bottom-up approach will be closer to the real situation with information on industry-specific default heterogeneity.

## 6. Empirical Results

### 6.1. Parameter Estimates

After employing the forward intensity model to estimate the original default forward intensity of all firms, we estimate $\beta_j(1)$ by maximizing the pseudo-likelihood function 17. Table 1 shows the estimated values of 10 industries' parameters.

**Table 1. Maximum pseudo-likelihood estimates for $\beta_j(1)$.**

| Industry | Basic Materials | Communications | Consumer (Cyc) | Consumer (N-Cyc) | Diversified |
|---|---|---|---|---|---|
| $\beta_j(1)$ | 775.41 | 873.63 | 1152.10 | 19,243.82 | 705.98 |
| Industry | Energy | Financial | Industrial | Technology | Utilities |
| $\beta_j(1)$ | 2423.68 | 276.79 | 1135.92 | 613.32 | 685.59 |

According to Formula (14), the larger $\beta_j(1)$ is, the closer posterior PDs are to prior PDs. This means that the firm in the $j$-th industry is slightly influenced by other firms' defaults in the same industry, and posterior probability does not change much. According to the estimated values of 10 industries' parameters, a firm in the consumer (non-cyclical) industry is least influenced by other firms' defaults in the same industry. In contrast, financial firms' defaults influence other financial firms the most. Firms in basic materials, diversified, technology, and utilities industries can also be influenced by other firms' defaults in the same industries.

After computing all $Z_j(m)$ and $TZ_j(m)$ values, we estimate $\gamma_j(l)$ for all horizons in all industries. Tables 2–5 show estimated values of $\gamma_{j1}(l), \gamma_{j2}(l), \gamma_{j3}(l), \gamma_{j4}(l)$ for some representative prediction horizons, and $\gamma_{j1}(l), \gamma_{j2}(l), \gamma_{j3}(l), \gamma_{j4}(l)$ are parameters of $Z_j, TZ_j, Z_{j,other}, TZ_{j,other}$.

**Table 2. Maximum pseudo-likelihood estimates for $\gamma_{j1}(l)$.**

| $Z_j$ | $\gamma_{j1}(1)$ | $\gamma_{j1}(2)$ | $\gamma_{j1}(3)$ | $\gamma_{j1}(6)$ | $\gamma_{j1}(12)$ | $\gamma_{j1}(24)$ | $\gamma_{j1}(36)$ |
|---|---|---|---|---|---|---|---|
| Basic Materials | 2.491185 | 2.280642 | 2.155384 | 1.899994 | −0.17849 | −1.35955 | −0.44241 |
| Communications | −2.33016 | −2.48579 | −2.58817 | −3.55322 | −1.46946 | −0.95426 | −1.068 |
| Consumer (Cyc) | −0.17068 | −0.3005 | −0.29659 | −1.6231 | 0.270778 | 0.515792 | 2.265838 |
| Consumer (NC) | −2.71041 | −2.76414 | −2.86428 | −2.44659 | −3.08668 | −1.99594 | −0.90574 |
| Diversified | −1.40276 | −0.98109 | −0.66264 | −1.03316 | −0.40614 | −0.40304 | −0.98643 |
| Energy | −4.25579 | −4.065 | −4.19237 | −3.32675 | −1.30519 | −0.38496 | 1.197159 |
| Financial | 2.734843 | 2.868124 | 2.859868 | 2.655078 | 2.034357 | −0.37494 | 0.082072 |
| Industrial | 2.854156 | 2.871496 | 2.946054 | 3.540954 | 3.536143 | 2.546084 | 0.027085 |
| Technology | −0.25124 | −0.24487 | 0.196717 | −0.99298 | −0.11297 | −1.51631 | −0.15389 |
| Utilities | −2.21956 | −2.17543 | −2.34536 | −2.69005 | −2.0572 | −2.72963 | −0.02748 |

**Table 3.** Maximum pseudo-likelihood estimates for $\gamma_{j2}(l)$.

| TZ$_j$ | $\gamma_{j2}(1)$ | $\gamma_{j2}(2)$ | $\gamma_{j2}(3)$ | $\gamma_{j2}(6)$ | $\gamma_{j2}(12)$ | $\gamma_{j2}(24)$ | $\gamma_{j2}(36)$ |
|---|---|---|---|---|---|---|---|
| Basic Materials | −2.24804 | −3.07666 | −2.29236 | −1.39137 | 0.774615 | 2.2439 | 0.465381 |
| Communications | 2.417619 | 3.533761 | 2.423067 | 4.272601 | −0.84387 | 0.828925 | 0.20722 |
| Consumer (Cyc) | 0.258211 | 0.999959 | 0.223138 | 0.895433 | 0.307311 | −0.60722 | −1.1084 |
| Consumer (NC) | −2.17874 | 2.099645 | 2.688754 | 2.547406 | 1.794337 | 9.730836 | 3.621596 |
| Diversified | −1.33008 | −1.43756 | 1.718504 | −0.99719 | 1.340722 | −1.39585 | −0.19119 |
| Energy | 0.945732 | 1.48091 | 3.675942 | 1.215448 | 3.31764 | 0.285514 | 2.258077 |
| Financial | −2.99554 | −2.91961 | −3.03768 | −2.8092 | −1.84164 | 0.348643 | 0.618982 |
| Industrial | −2.59238 | −2.71164 | −2.95667 | −3.20924 | −3.50535 | −2.62851 | −0.19389 |
| Technology | 0.411408 | −0.05615 | −0.10175 | 1.319924 | 0.447983 | 0.477757 | 0.030217 |
| Utilities | 1.668564 | 3.580656 | −0.04558 | 0.004809 | 2.576682 | −0.74262 | 0.720997 |

**Table 4.** Maximum pseudo-likelihood estimates for $\gamma_{j3}(l)$.

| Z$_{j,other}$ | $\gamma_{j3}(1)$ | $\gamma_{j3}(2)$ | $\gamma_{j3}(3)$ | $\gamma_{j3}(6)$ | $\gamma_{j3}(12)$ | $\gamma_{j3}(24)$ | $\gamma_{j3}(36)$ |
|---|---|---|---|---|---|---|---|
| Basic Materials | −2.52744 | −2.34069 | −2.21324 | −2.02293 | 0.116848 | 1.208248 | 0.370522 |
| Communications | 2.736875 | 2.871824 | 2.979698 | 3.771385 | 1.623522 | 1.195812 | 1.250113 |
| Consumer (Cyc) | 0.086383 | 0.211614 | 0.229138 | 1.523775 | −0.33232 | −0.54249 | −2.31581 |
| Consumer (NC) | 2.8266 | 2.876228 | 2.993546 | 2.629115 | 3.242611 | 2.099184 | 0.765712 |
| Diversified | 2.039327 | 1.601927 | 1.279474 | 1.789817 | 1.038678 | 0.736229 | 1.258517 |
| Energy | 4.680227 | 4.462585 | 4.607325 | 3.661166 | 1.546356 | 0.606136 | −1.10992 |
| Financial | −2.80724 | −2.94792 | −2.92779 | −2.76668 | −2.16542 | 0.132517 | −0.26506 |
| Industrial | −2.80509 | −2.84218 | −2.90324 | −3.50324 | −3.48343 | −2.55798 | −0.07193 |
| Technology | 0.568141 | 0.597914 | 0.083002 | 1.318448 | 0.537657 | 1.79763 | 0.146919 |
| Utilities | 1.677209 | 1.614704 | 1.666923 | 1.967737 | 1.404774 | 2.075835 | −0.47442 |

**Table 5.** Maximum pseudo-likelihood estimates for $\gamma_{j4}(l)$.

| TZ$_{j,other}$ | $\gamma_{j4}(1)$ | $\gamma_{j4}(2)$ | $\gamma_{j4}(3)$ | $\gamma_{j4}(6)$ | $\gamma_{j4}(12)$ | $\gamma_{j4}(24)$ | $\gamma_{j4}(36)$ |
|---|---|---|---|---|---|---|---|
| Basic Materials | 2.147868 | 3.726177 | 2.787542 | 1.373924 | 0.439178 | −1.03204 | 1.224042 |
| Communications | −2.72153 | −2.59917 | −2.4975 | −2.92083 | 0.923662 | −1.47403 | 0.027272 |
| Consumer (Cyc) | −0.63154 | −0.00092 | −0.08518 | −1.26268 | 1.590467 | 2.310409 | 2.734578 |
| Consumer (NC) | −2.89664 | −2.00488 | −2.07615 | −2.74982 | −2.17565 | −1.41085 | 0.23462 |
| Diversified | −0.90503 | −1.67394 | 0.751807 | −1.69593 | 1.085887 | 2.508569 | 1.608284 |
| Energy | −3.07229 | −2.68038 | −4.0168 | −5.10284 | 0.382157 | 1.557167 | 1.929201 |
| Financial | 4.367602 | 4.145667 | 2.898224 | 0.882199 | 3.998458 | 0.975475 | 0.269972 |
| Industrial | 3.307187 | 3.975671 | 3.591346 | 2.965927 | 4.566214 | 2.623784 | 0.640499 |
| Technology | 1.176082 | 0.753229 | 1.602314 | −1.25978 | 0.404673 | 1.77964 | 2.752035 |
| Utilities | −2.04431 | −1.27136 | −2.15938 | −3.01697 | −0.00345 | −1.59101 | −0.13897 |

Table 2 reflects the impact of current values of industry-specific default heterogeneity indicators in 10 industries for some representative horizons. Table 3 reflects the impact of trends in default heterogeneity indicators in 10 industries for some representative horizons. In the industrial industry, the impact of co-movements is largest and longest. Default clustering may influence the industrial industry over more than 2 years. In the financial industry, the impact of co-movements is also large and lasts for more than a year.

Table 4 reflects the impact of the current values of industry-specific default hetero-geneity indicators in other industries for some representative horizons. Table 5 reflects the impact of trends in industry-specific default heterogeneity in other industries for some representative horizons. Some industry categories, like communications, consumer (non-cyclical), diversified, and energy are influenced much more by co-movements in other industries' defaults than by themselves.

*6.2. Comparing with the Number of Defaults*

Since the forward intensity model is practical for estimating multi-period PDs with term structure, we chose to extend it by constructing industry-specific default heterogeneity indicators. Therefore, we need to compare the original forward intensity model with our extended model. In addition, neural networks and machine learning are also good ways to improve default prediction. Previous research on machine learning usually only predicts defaults through discrimination or calculation of credit scores rather than estimation of accumulative PDs for multiple periods. For example, Barboza et al. [25] predicted bankruptcy one year prior to the event and compared the performance of different machine learning models, including support vector machines, bagging, boosting, and random forest. Gunnarsson et al. [26] constructed a multilayer perceptron network and a deep belief network and compared their performance for credit scoring. However, with the development of PD models, it is possible to predict multi-period PDs through machine learning. For example, Sigrist and Leuenberger [13] combined econometric models with different machine learning models to estimate multi-period cumulative PDs and found that tree-boosting has the highest prediction accuracy. In this paper, we need to judge whether our industry default heterogeneity indicators are helpful to improve default-predicting ability for Chinese-listed firms, and the comparison between the original forward intensity model and our extended model is as follows.

Comparing the aggregated number of defaults with aggregated PDs is widely used to evaluate credit risk model performance. For each horizon, we computed the aggregate accumulated PDs of all surviving firms using data for the first day of the month and aggregated numbers of defaults in the prediction horizons for the same firms. Then, we compared them for all prediction horizons. Figures 3 and 4 show a comparison of the realized number of defaults, original aggregated accumulated PDs, and new aggregated accumulated PDs for a 12-month prediction horizon in sample and out of sample, respectively.

**Figure 3.** This figure shows a comparison of the realized number of defaults, original aggregated accumulated PDs, and new aggregated accumulated PDs for a 12-month prediction horizon in the sample.

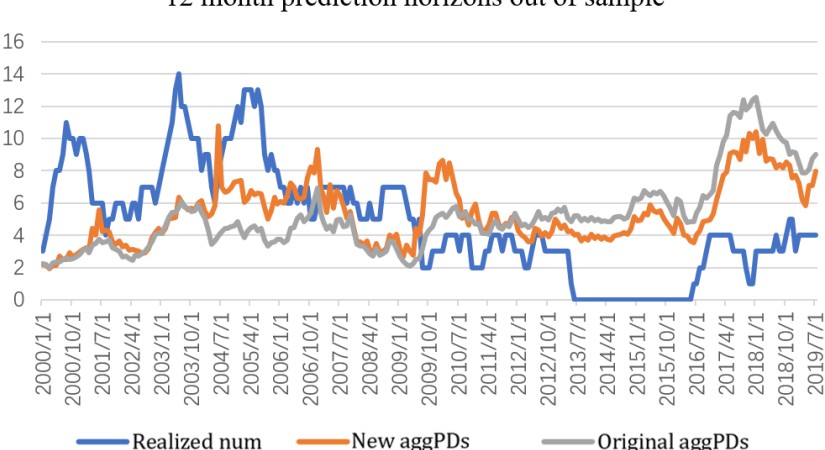

**Figure 4.** This figure shows a comparison of the realized number of defaults, original aggregated accumulated PDs, and new aggregated accumulated PDs for a 12-month prediction horizon out of sample.

Obviously, new aggregated PDs that consider industry-specific default heterogeneity are closer to the realized number both in and out of sample. From 2003 to 2005, a large number of firms defaulted, and new aggregated PDs that consider industry-specific default heterogeneity are higher than the original PDs for the prediction horizon of 12 months. In contrast, the new aggregated PDs are lower than the original PDs for the period 2013 to 2015. This indicates that our model can capture more effective information on industry-specific default heterogeneity and its impact.

*6.3. Prediction Accuracy Ratio*

Accuracy ratio (AR) is widely adopted to evaluate the predictive ability of credit risk models. AR can reflect the effective information on defaults in the future that a credit risk model contains. If the AR is zero, it means the model is a zero-information model. If the cumulative accuracy profile is at the 45° line, then the model is a zero-information model. The higher the AR, the better the predictive power of the model, and the cumulative accuracy profile will be further above the 45° line. Readers may refer to Vassalou and Xing [27] for more details. The AR they obtained for Merton's model is 0.592, which means the model contains substantial, effective information on defaults in the future. Figures 5 and 6 show the cumulative accuracy profiles for calibrated PDs from January 2000 to December 2019 for horizons of 1, 2, 3, 6, 12, 24, and 36 months in and out of sample, respectively. The model's cumulative accuracy profiles are obviously above the 45° line for all prediction horizons. In particular, the predictive power of the model is strong for horizons of no more than 1 year.

Figures 7 and 8 contrast the ARs of new PDs with the original PDs for all horizons in and out of sample. The ARs of new PDs are always higher than those of original PDs, both in sample and out of sample, for all horizons. On the other hand, the out-of-sample ARs of the new PDs are not lower than the in-sample ARs. For the short-term prediction horizons, the ARs of our new PDs were greatly improved compared with the original PDs out of sample. When the prediction horizons increase, our calibrated PDs improve less. However, when the prediction horizon increases to 3 years, the ARs of our new PDs are still higher than for the original PDs.

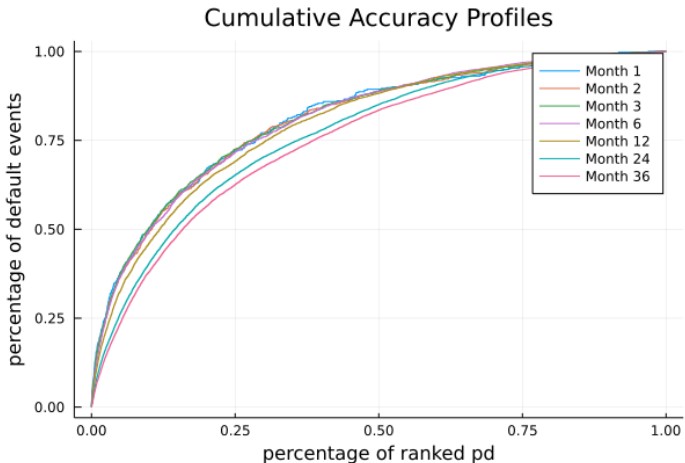

**Figure 5.** This figure shows the in-sample cumulative accuracy profiles of new PDs from January 2000 to December 2019 for different prediction horizons.

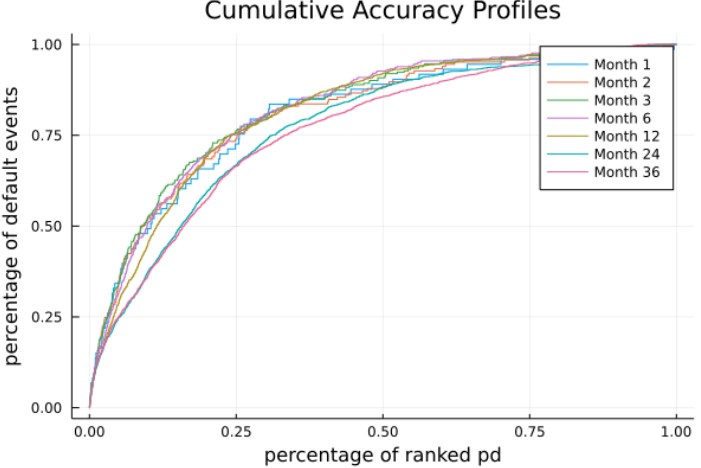

**Figure 6.** This figure shows the out-of-sample cumulative accuracy profiles of new PDs from January 2000 to December 2019 for different prediction horizons.

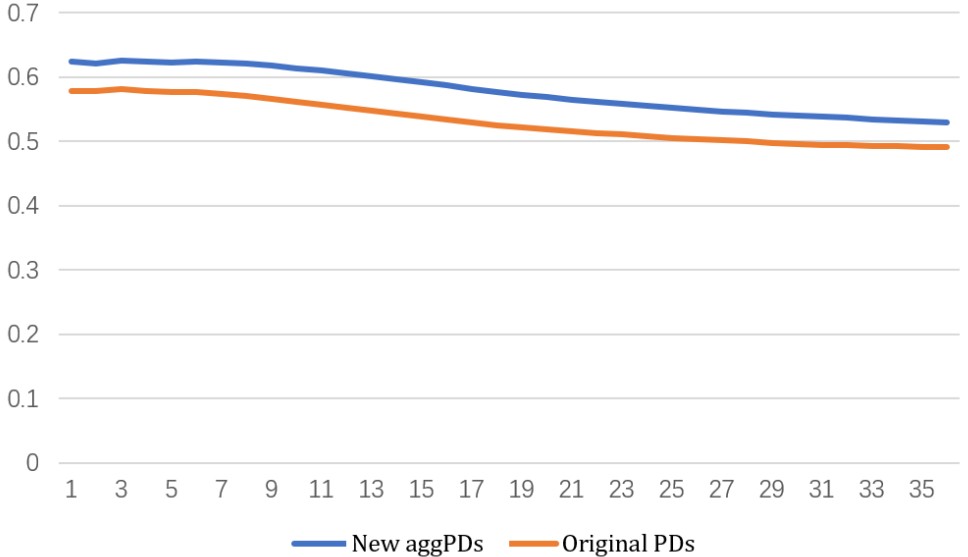

**Figure 7.** This figure compares the in-sample cumulative accuracy profiles of new PDs and original PDs for horizons of 1–36 months.

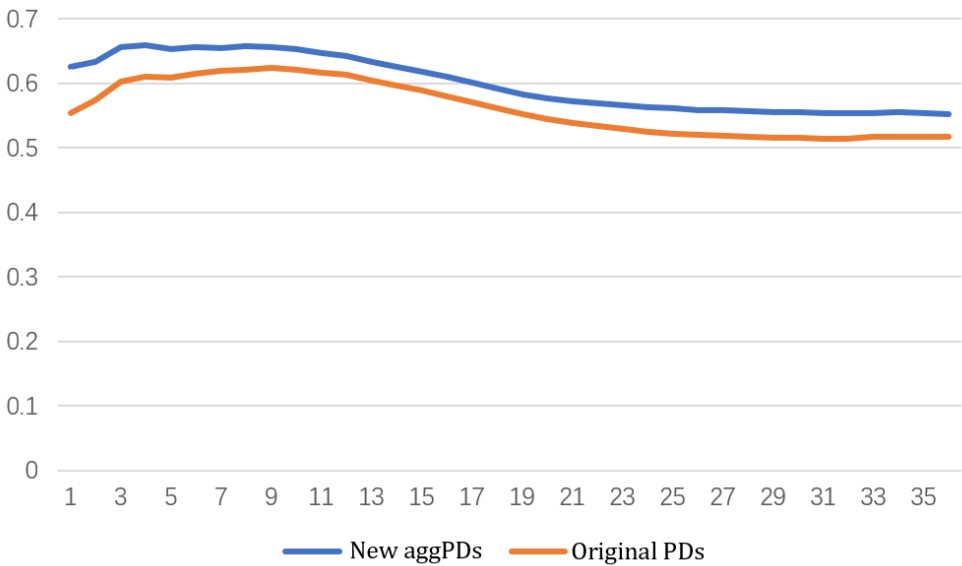

**Figure 8.** This figure compares the out-of-sample cumulative accuracy profiles of new PDs and original PDs for horizons of 1–36 months.

Tables 6 and 7 show the values of the ARs for horizons of 1, 2, 3, 6, 12, 24, and 36 months in and out of sample, respectively. The ARs of our new PDs in the sample show a similar increase, of about 7%. Out of sample, the ARs of our new PDs are relatively increased by 8% when the prediction horizons are no more than 3 months. When the prediction horizons increase to more than 6 months, the relative increase in ARs is stable at around 6%. This means that our model is also helpful for long-horizon prediction.

**Table 6.** Comparison of the ARs in the sample.

| Accuracy Ratio | 1 Month | 2 Months | 3 Months | 6 Months | 12 Months | 24 Months | 36 Months |
|---|---|---|---|---|---|---|---|
| Original PDs | 0.5785 | 0.5782 | 0.5811 | 0.5772 | 0.5520 | 0.5083 | 0.4913 |
| Revised PDs | 0.6246 | 0.6202 | 0.6255 | 0.6238 | 0.6056 | 0.5554 | 0.5293 |
| Increased (%) | 7.4 | 6.8 | 7.1 | 7.5 | 8.8 | 8.5 | 7.2 |

**Table 7.** Comparison of the ARs out of sample.

| Accuracy Ratio | 1 Month | 2 Months | 3 Months | 6 Months | 12 Months | 24 Months | 36 Months |
|---|---|---|---|---|---|---|---|
| Original PDs | 0.5543 | 0.5733 | 0.6032 | 0.6149 | 0.6131 | 0.5254 | 0.5174 |
| Revised PDs | 0.6262 | 0.6335 | 0.6559 | 0.6557 | 0.6418 | 0.5628 | 0.5527 |
| Increased (%) | 11.5 | 9.5 | 8 | 6.2 | 4.8 | 6.6 | 6.4 |

## 7. Conclusions

All in all, industry-specific default heterogeneity is a problem that cannot be ignored in default prediction for a large portfolio. To address this issue, we extended the forward intensity model and introduced industry-specific default heterogeneity indicators into the model. We divided Chinese-listed firms into 10 industries and estimated the parameters of industry-specific default heterogeneity indicator functions for different industries. Then, we calculated the default heterogeneity indicators for all industries for each month and maximized the pseudo-likelihood function of new PDs. We measured the impacts that all firms received from within- and across-industry default heterogeneity for all horizons. Finally, we computed new PDs containing information about co-movements of default within and across industries over time. The new PDs improve the ARs and reduce the

gap between aggregated PDs and the realized number of defaults, both in and out of sample, for all horizons. Through theoretical modeling and data analysis, we reached the following conclusions:

(1)    Co-movements in different industries are very heterogeneous. Defaults in some industries are greatly affected by co-movements of defaults within the industry, and some industries are greatly affected by co-movements of default in other industries.

(2)    Extending the forward intensity model, we studied how firms' defaults are influenced by co-movements of default within the industry and across industries. We computed all Chinese-listed firms' PDs by measuring the influence of co-movements of default within the industry and across industries.

(3)    The empirical results show that new PDs considering the impact of industry-specific default heterogeneity within and across industries have stronger predictive ability.

For both in-sample and out-of-sample predictions, new PDs' ARs improved by more than 6% relative to all prediction horizons. Out of sample, short-term PDs' ARs improved by up to 8% or more. We also compared the aggregated PDs and the realized number of defaults for one year and found that new PDs have a smaller gap from the realized number of defaults. The main contribution of this paper is that we extended the forward intensity model to make PDs contain information about default heterogeneity in industries. In addition, we measured the impact of default heterogeneity within and across industries. This makes the model capture more co-movements of defaults in the industry and the region. This is of significance not only for individual investors to avoid risk but also for credit risk supervision departments to detect systemic credit risk. For examples, it is helpful for a stress test to capture risk spillover within and across industries when default clustering occurs.

In addition, we find that the levels and trends of systemic credit risk across the region have a very different impact on various industries. The credit risk of industrial and financial firms, for example, is more sensitive to the increasing frequency of defaults across the region. While the cluster of defaults across the region is accelerating rapidly, even if it has not yet reached a high level, regulators must pay attention to these two industries. When the default cluster is already severe, diversified, energy, and communications firms are more likely to default. The non-cyclical consumption sector is more affected by external influences than the cyclical consumption sector. In a bad economic climate, regulators can focus on limiting leverage in these sectors to avoid a chain reaction. In addition, industries with large fluctuations in the industry-specific default heterogeneity indicator are more affected by systemic credit risk, such as the financial industry. When a default cluster appears, the financial industry's overall credit risk rises much higher than in other industries. When economic environment is good, the financial industry's overall credit risk declines more than in other industries. For industries with small fluctuations in the industry-specific default heterogeneity indicator, credit risk is mainly affected by the firm-specific attributes and observed common factors, and the new PD is closer to the original PD. Therefore, regulators could use such a model to do stress testing on the whole region and control the leverage of firms in financial, diversified and energy industries according to the level of systemic credit risk.

Credit risk modeling for large portfolios still faces challenges. Future research has several directions. In order to maintain stability, the spillover credit risks from other industries which we did not classify in this paper could be considered. Researchers can use machine learning methods to study the relationships between risk spillovers across industries. On the other hand, Duan et al. [1] adopted the exponential function to fit the firm's forward default intensity in the forward intensity approach. Compared with the linear function, the exponential function has more advantages. Researchers can also try to use the neural network method to fit the forward default intensity and realize a combination of the machine learning model with the forward default intensity model. Researchers can also conduct a more detailed study on how defaults spread within and across industries. We need deeper study on sources of default heterogeneity in industries,

and it is important to improve credit risk models. Credit risk models which capture more unobserved co-movements of defaults in industries will be meaningful for improving credit risk models.

**Author Contributions:** Conceptualization, Z.N.; methodology, Z.N.; software, Z.N.; validation, Z.N.; formal analysis, Z.N.; investigation, Z.N.; resources, M.J.; writing—original draft preparation, Z.N. and W.Z.; writing—review and editing, Z.N. and W.Z.; supervision, M.J.; project administration, M.J.; funding acquisition, M.J. All authors have read and agreed to the published version of the manuscript.

**Funding:** This research was supported in part by the National Natural Science Foundation of China Grant No. 71831005 and No. 71502044. Funder: Minghui Jiang.

**Data Availability Statement:** Our data source is the Credit Research Initiative (CRI) database of the National University of Singapore.

**Conflicts of Interest:** The authors declare no conflict of interest.

## Appendix A

In this paper, we employed the gradient descent method to maximize the pseudo log-likelihood function and estimate $\beta_j(1)$. Appendix A will show the pseudo log-likelihood function of $\beta_j(1)$ and its gradient.

We substitute Formula (14) into Formula (18) and take the log of it:

$$L_{l=1}^{\text{posterior}} = \prod_{m=1}^{M} \prod_{i=1}^{I_j(m)} ln\hat{P}_{j,l=1}\left(\beta_j(1), \tau_i, \overline{\tau}_i, h_i(m,m)\right) = \prod_{j=1}^{J} L_{j,l=1}^{\text{posterior}}. \tag{A1}$$

Here, $L_{l=1}^{\text{posterior}}$ can be decomposed by industry, and $m = 1$ denotes the first month from which we want to compute new PDs. Note that we also have data before this month, so we can also calculate $Z_j(m)$ and $TZ_j(m)$ when $m < 1$. If we maximize $j$ sets of $L_{j,l=1}^{\text{posterior}}$, $L_{l=1}^{\text{posterior}}$ will be the maximum. Let $h_i^E(m,n)$ be the default intensity, on the condition that the firm has this kind of event at $n$-th month:

$$h_i^E(m,n) = 1_{\{E_i(n)=E\}} \times h_i(m,n), \tag{A2}$$

where $E$ is the event type, which can be 0, 1, or 2 to denote survival, default, or other exit respectively. Here, $E_i(n)$ is the event type of the $i$-th firm in the $n$-th month and $L_{j,l=1}^{\beta,\text{posterior}}$ is the decomposed part of $L_{j,l=1}^{\text{posterior}}$, which is only associated with $\beta_j(1)$. Then, it can be expressed as follows:

$$\begin{aligned}
L_{j,l=1}^{\beta,\text{posterior}} = &-\tau \sum_{m=1}^{M} \sum_{i=1}^{I_j(m)} \frac{\beta_j(1)+I_j(m-1)\frac{y_j(m-1,m-1)}{I_j(m)}}{\tau\sum_{i=1}^{j} h_i(m-1,m-1)} \left(h_i^0(m,m)+h_i^2(m,m)\right) \\
&+ \sum_{m=1}^{M} \sum_{i=1}^{I_j(m)} ln\left\{ 1 - \exp\left( -\tau \frac{\beta_j(1)+I_j(m-1)\frac{y_j(m-1,m-1)}{I_j(m)}}{\tau\sum_{i=1}^{j} h_i(m-1,m-1)} h_i^1(m,m) \right) \right\}
\end{aligned} \tag{A3}$$

The gradient of $L_{j,l=1}^{\beta,\text{posterior}}$ is expressed as

$$
G_{j,l=1}^{\beta,\text{posterior}} = -\tau \sum_{m=1}^{M} \sum_{i=1}^{I_j(m)} \frac{I_j(m-1)-I_j(m-1)\frac{y_j(m-1,m-1)}{\tau\sum_{i=1}^{I_j(m)} h_i(m-1,m-1)}}{\left(\beta_j(1)+I_j(m-1)\right)^2} \left(h_i^0(m,m)+h_i^2(m,m)\right)
$$

$$
+\tau \sum_{m=1}^{M} \sum_{i=1}^{I_j(m)} \frac{\exp\left(-\tau\frac{\beta_j(1)+I_j(m-1)\frac{y_j(m-1,m-1)}{I_j(m)}}{\beta_j(1)+I_j(m-1)} h_i^1(m,m)\right)}{1-\exp\left(-\tau\frac{\beta_j(1)+I_j(m-1)\frac{y_j(m-1,m-1)}{I_j(m)}}{\beta_j(1)+I_j(m-1)} h_i^1(m,m)\right)}
$$

$$
\times \frac{I_j(m-1)-I_j(m-1)\frac{y_j(m-1,m-1)}{I_j(m)}}{\left(\beta_j(1)+I_j(m-1)\right)^2} h_i^1(m,m). \tag{A4}
$$

Similarly, we also calculate the log of the pseudo likelihood functions of $\gamma_j(l)$ and then calculate the gradient. The pseudo log-likelihood functions of $\gamma_j$ can be expressed as:

$$
L_l = \prod_{l=1}^{l_{max}} \prod_{m=1}^{M-l+1} \prod_{i=1}^{I(m)} ln\hat{P}_{j,l}\left(\gamma_{j1},\gamma_{j2},\gamma_{j3},\gamma_{j4},\tau_i,\overline{\tau}_i,X_i(m),\,h_i(m,m+l-1)\right) = \prod_{l=1}^{l_{max}} \prod_{j=1}^{J} L_{j,l}, \tag{A5}
$$

where $l_{max}$ is the largest prediction horizon we set and $L_l$ can be decomposed by industry and prediction horizon. Then, $L_{j,l}^{\gamma}$ is the decomposed part of $L_{j,l}$ which is only associated with $\gamma_j(l)$, and can be expressed as follows:

$$
L_{j,l}^{\gamma} = -\tau \sum_{m=1}^{M} \sum_{i=1}^{I_j(m)} \left\{\exp\left[X_j(m)\gamma_j(l)\right]\left[\left(h_i^0(m,m+l-1)+h_i^2(m,m+l-1)\right)\right]\right\}
$$

$$
+ \sum_{m=1}^{M} \sum_{i=1}^{I_j(m)} \ln\left\{1-\exp\left\{-\tau\cdot\exp\left[X_j(m)\gamma_j(l)\right]h_i^1(m,m+l-1)\right\}\right\}, \tag{A6}
$$

Let $G_{j,l}^{\gamma}$ be the gradient of $L_{j,l}^{\gamma}$:

$$
G_{j,l}^{\gamma} = -\tau \sum_{m=1}^{M} \sum_{i=1}^{I_j^{(m)}} X_j(m)^T \exp\left[X_j(m)\gamma_j(l)\right]\left[h_i^0(m,m+l-1)+h_i^2(m,m+l-1)\right]
$$

$$
+\tau \sum_{m=1}^{M} \sum_{i=1}^{I_j^{(m)}} \frac{\exp\left\{-\tau\cdot\exp\left[X_j(m)\gamma_j(l)\right]h_i^1(m,m+l-1)\right\}}{1-\exp\left\{-\tau\cdot\exp\left[X_j(m)\gamma_j(l)\right]h_i^1(m,m+l-1)\right\}}
$$

$$
\times X_j(m)^T \exp\left[X_j(m)\gamma_j(l)\right]\left(h_i^1(m,m+l-1)\right). \tag{A7}
$$

The gradient descent method is used to estimate $\beta$ and $\gamma$. Then, $L_{j,l=1}^{\beta}$ and $L_{j,l}^{\gamma}$ will be the maximum. Finally, we realize the estimation on the pseudo likelihood function and get the value of all parameters.

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
