# Peer review of "Default Prediction with Industry-Specific Default Heterogeneity Indicators Based on the Forward Intensity Model"

_axioms, doi:10.3390/axioms12040402_

Round 1

Reviewer 1 Report

Recommendations for Manuscript axioms-2301599Default Prediction with Industry-specific Default Heterogeneity Indicators Based on the Forward Intensity Model/29.03.2023 for the Axioms Journal.

General Comments

From my point of view, it is a very interesting topic and simultaneously it seems that to the best of my knowledge is the first empirical study extends the forward intensity model to predict defaults of Chinese listed-firms with the information about default heterogeneity in industries. The authors construct industry-specific the default heterogeneity indicator which is helpful to improve the model performance especially for predicting defaults of a large portfolio, which is of significance for credit risk management in China or other regions.

The paper contains the following sections: Introduction, Literature review, Compute PDs by Industry-Specific Heterogeneity Indicators, Pseudo-likelihood functions, Data and preliminary Analysis, Empirical Results,   and Conclusions.

However, I find some recommendations:

1.      The abstract must contain the main purpose of the paper, the research method used in the research and the main contributions.

2.      It would be very useful to add in the "Introduction" section the purpose, objectives and hypothesis of the research. I consider that a weak point of the paper is that the authors did not show the novelty of the paper compared to other works. That is why, I consider that the introduction should specify the novelty of the paper compared to other papers published in this area.

3.      The Conclusions part of the paper must be expanded with Implications and Limitations of the results of the paper.

4.      The research is well based on science and the results are in agreement with the theoretical part. From my point of view, the paper is original and the topic addressed brings added value to the specialized literature regarding to predict defaults of Chinese listed-firms with the information about default heterogeneity in industries. The paper is well written and easy to read.

5.      At the same time, I consider that the conclusion part of the paper should be expanded.

6.      The research is well based on science and the results are in agreement with the theoretical part. The  model applied to the analyzed data is correctly used in the analysis undertaken, it is a strength point of this paper. 

7.       Since the authors in the construction of this indicator are based on data from several industries, we recommend citing the following papers published in MDPI journals:

1.      Batrancea L.M. (2021) An Econometric Approach on Performance, Assets, and Liabilities in a Sample of Banks from Europe, Israel, United States of America, and Canada. Mathematics, 9(24):3178. https://doi.org/10.3390/math9243178.

8.      The conclusions at the end of the paper should be expanded showing the economic policy implications of the research results.

In conclusion, the article should be improve. It should also be enhanced with a review of the literature adequate to the subject and a broader interpretation and commentary of the research results.

Reviewer 2 Report

The article discusses an approach to assessing credit risk, taking into account the specific industry characteristics of firms. The construction of metrics for risk assessment is based internal factors specific to the functioning of firms, as well as environmental factors - sectoral, institutional under their heterogeneity and non-linearity. This is what distinguishes this work from others.

The article as a whole complies with the requirements of the journal and should be published.

However, some minor adjustments need to be made:

1. Authors could indicate in the work the possibility of using neural network tools and machine learning methods. Compare the results of the authors with the others results that predict the probability of default using artificial intelligence methods.

2. How will the proposed model take into account low-probability risks and their consequences?

3. Some expansion of the references is possible, taking into account the works using the methods indicated in comment 1.

4. Some pictures are hard to read (Fig. 1,2) due to not very successful color palette of the lines. The lines are hard to distinguish from each other
